# It's Not a Modality Gap: Characterizing and Addressing the Contrastive Gap

## Abstract

Learning jointly from images and texts using contrastive pre-training has emerged as an effective method to train large-scale models with a strong grasp of semantic image concepts. For instance, CLIP (Radford et al., 2021), pre-trained on a large corpus of web data, excels in tasks like zero-shot image classification, object detection, geolocalization, and more. These contrastive models embed input images and texts into a shared representational space. Recently, it was claimed that models like CLIP show a *modality gap*, where image and text embeddings occupy disjoint areas in the representational space. Previous studies attribute this gap to factors like data artifacts (mismatched pairs), model architecture artifacts (the cone effect), and the nature of the loss landscape (getting stuck in local minima). We demonstrate that, even after accounting for these factors, and even when using the *same modality*, the contrastive loss actually *creates* a gap during training. As a result, we propose renaming this phenomenon the *contrastive gap*. We show that the contrastive gap is exacerbated by training with small batch sizes in high-dimensional spaces, causing embeddings of each modality to occupy small disjoint portions of the latent space. Our experiments show that minimizing the contrastive gap via the addition of uniformity and alignment terms optimizes the representational space and conveys better performance on downstream tasks such as image-text retrieval, zero-shot image classification, and multi-modal arithmetic.

## 1 Introduction

Multi-modal models map inputs from different modalities into a unified representational space, such that semantically similar inputs from different modalities map to nearby points in the representational space. This can be practically useful because it allows transfer between modalities, but it is also more consistent with the human sensory experience, which collects information across many modalities to better understand the world. In the context of learning from paired images and text, CLIP (Contrastive Language Image Pre-training) (Radford et al., 2021) establishes a strong proof-of-concept for this reasoning. CLIP's multi-modal contrastive loss allows the model to predict the text associated with an image and vice versa. CLIP scales this approach to a very large dataset of 400M image-caption pairs, learning embeddings that cover a wide variety of visual concepts applicable to many downstream tasks.

But, while CLIP is powerful, it suffers from a *modality gap* (Liang et al., 2022), wherein image embeddings reside in a space disjoint from that of the text embeddings. This phenomenon is also seen in multi-modal contrastive models in other domains such as medical images (Zhang et al., 2021), videos (Xu et al., 2021), amino acid sequencing (`https://github.com/MicPie/clasp`) and brain decoding (Luo et al., 2024). Prior work has shown that performance on downstream tasks improves when we minimize this gap, which can be achieved by simply shifting one modality's embeddings in CLIP space to change the size of the gap (Liang et al., 2022) through projection (transforming the embeddings through some projection operation) (Zhou et al., 2023), or through fine-tuning (Oh et al., 2023). Recent work has additionally shown that learning latent spaces of visual embeddings via alignment with human similarity judgments improves downstream task performance (Muttenthaler et al., 2023). Similarly, work relating CLIP embeddings to human brain data found that representations that had reduced the gap between modalities also led to improved downstream model performance (Luo et al., 2024). Therefore, analyzing and closing this gap is a promising direction to improve upon the strong representational capacity of CLIP and its variants.

In this paper, we conducted a comprehensive study of what causes the *modality gap* and simple ways to close it. We make the following contributions:

- **It's not a modality gap.** After summarizing the common purported causes of the modality gap, we perform comprehensive experiments that show that accounting for these factors does *not* close the gap, suggesting that the present understanding of modality gap may be flawed.
- **It's a *contrastive* gap.** We present experiments that demonstrate that the gap is a byproduct of a high dimensional CLIP space, combined with contrastive loss that encourages CLIP embeddings to occupy a lower dimensional manifold relative to the latent space.
- **The contrastive gap can be closed.** We show that simply fine-tuning CLIP by adding a factor for uniformity and alignment can reduce the size of the gap by distributing the embeddings more uniformly throughout CLIP's latent space.
- **Closing the contrastive gap improves downstream performance.** Finally, we present experiments to show that closing the contrastive gap, and thereby creating more aligned and uniformly distributed representations, creates a representational space that is better for downstream tasks, including image-text retrieval, zero-shot image classification, and multi-modal embedding arithmetic.

## 2 BACKGROUND

A multi-modal contrastive model learns a representational space in which semantically paired samples from different modalities (*positive pairs*, e.g., an image and its associated caption) are closer together in the latent space compared to other randomly chosen pairs (*negative pairs*, e.g., an image and the caption associated with another randomly chosen image). Contrastive learning was originally developed for the unimodal setting to learn self-supervised representations. One such popular model is SimCLR (Chen et al., 2020), which uses the NT-Xent (Normalized Temperature-scaled Cross entropy) loss to efficiently learn robust image representations. The NT-Xent loss is different from previous contrastive learning methods as it does not need explicit negative samples. The NT-Xent loss works by treating samples other than the positive pair in a training mini-batch as soft-negatives, pushing them away from the positives. Unlike previous contrastive methods, the NT-Xent loss also normalizes the embeddings to lie on a unit hypersphere in the representational space.

Wang & Isola (2020) introduced *uniformity* and *alignment* factors as desirable properties of the representational space and showed that the NT-Xent loss optimizes the uniformity and alignment properties in the limit of infinite negative samples, which is equivalent to infinite batch size for NT-Xent loss.

Many models have been recently proposed that generalize the contrastive loss to the multi-modal setting (Zhang et al., 2021; Jia et al., 2021; Radford et al., 2021). Before these, several works have explored the integration of different modalities using contrastive and triplet losses. Wang et al. (2018) uses a two-branch neural network architecture with a bidirectional retrieval ranking loss to learn joint embeddings for images and text. Wang et al. (2018) also included neighborhood-preserving constraints, sampling mini-batches such that they contained multiple positive matches for a given target, in order to properly capture the many-to-many relationships between images and texts (i.e multiple images matching one text sequence or vice versa). Mahajan et al. (2019) employs joint Wasserstein autoencoders to align multi-modal embeddings in a shared latent space. Further, Lee et al. (2018) proposes a stacked cross-attention mechanism to discover fine-grained correspondences between image regions and words for more accurate image-text matching. However, CLIP stands out from these approaches due to its simplicity and scalability. While prior work requires complex architectures and sampling strategies, CLIP uses a simple contrastive loss and random mini-batch sampling from a vast dataset (400M image-text pairs). CLIP's approach circumvents the requirement for complex architectures and additional loss terms, yet achieves strong zero-shot results on many standard image benchmarks. CLIP's efficiency and ability to generalize across tasks have led to its widespread adoption across a wide range of applications.

While CLIP has demonstrated remarkable advances in zero-shot learning and cross-modal tasks, Liang et al. (2022) observed that a *modality* gap appears in many models that use multi-modal contrastive learning. There have since been several attempts to close the gap while monitoring its effects on downstream task performance. Liang et al. (2022) show that altering the gap by simply trans-

lating the embeddings of one modality onto the other can positively impact zero-shot classification performance. Further, Oh et al. (2023) attribute the modality gap to the absence of hard-negatives in a training mini-batch and synthetically generated hard-negative samples from existing data points, showing that training with the hard-negatives improves representational quality. In this work, we frame the gap between the multi-modal embeddings as a problem of low uniformity and show that by simply optimizing for more alignment and uniformity (properties known to be desirable in uni-modal contrastive learning), we can significantly reduce the size of the gap, while increasing the quality of the representations learned.

## 2.1 CONTRASTING IMAGES AND TEXT USING CLIP LOSS

CLIP is an example of a contrastive model that learns image and text embeddings. CLIP loss ($L_{\text{CLIP}}$) is based on the *multi-class N-pair loss* that was first introduced in Sohn (2016). This loss has been termed as the NT-Xent loss (Chen et al., 2020) and has also been used in prior works (Wu et al., 2018; van den Oord et al., 2019). While NT-Xent loss is designed to work on data points from a single modality, $L_{\text{CLIP}}$ is adapted to work on two different modalities of data.

In our scenario, the multi-modal dataset contains $N$ images and corresponding captions. We obtain image embeddings $E_j^I \in \mathbb{R}^d$ by passing image $I_j$ through the image encoder. Similarly, we produce the text embedding $E_j^T \in \mathbb{R}^d$ by passing caption $T_j$ through the text encoder. CLIP aims to bring image embeddings and their corresponding caption embeddings closer together in CLIP latent space (*CLIP space*) by increasing the similarity (inner product $\langle ., . \rangle$) between the corresponding embeddings. The image and text embeddings are normalized to lie on a unit hypersphere in $\mathbb{R}^d$.

The full CLIP loss is:

$$L_{\text{CLIP}} = -\frac{1}{2N} \sum_{j=1}^{N} \log \left[ \frac{\exp\left(\langle E_j^I, E_j^T \rangle / \tau\right)}{\sum_{k=1}^{N} \exp\left(\langle E_j^I, E_k^T \rangle / \tau\right)} \right] - \frac{1}{2N} \sum_{k=1}^{N} \log \left[ \frac{\exp\left(\langle E_k^I, E_k^T \rangle / \tau\right)}{\sum_{j=1}^{N} \exp\left(\langle E_j^I, E_k^T \rangle / \tau\right)} \right]$$

$$(1)$$

Where the left term **contrasts images with the texts** ($\sum_{k=1}^{N}$ in the denominator loops over text embeddings as negatives for the $j$th image) and the right term **contrasts texts to images** ($\sum_{j=1}^{N}$ in the denominator loops over image embeddings as negatives for the $k$th text). $\tau$ represents the temperature parameter ($\tau = 0.01$ at the end of CLIP pre-training).

## 3 THE GAP IN MULTI-MODAL CONTRASTIVE LEARNING

Though CLIP effectively associates images with related texts, it also creates representational spaces with a *modality gap*, a phenomenon that has generated some interest. Liang et al. (2022) attributes this modality gap to two independent factors. First, they describe **the cone effect** of deep neural networks, in which different random initializations cause the embeddings of two encoders to occupy two non-overlapping narrow cones in CLIP space. Second, they note the existence of **mismatched pairs** in the dataset, where the pairing of images and captions is incorrect for some data points. Shi et al. (2023) studied the gap in three dimensions and attributed it to conflicting uniformity and alignment terms in the contrastive loss, claiming that this leads to the existence of local minima that encourage the gap. Further, (Oh et al., 2023; Liang et al., 2022) show that the modality gap persists even after fine-tuning. In our first experiment in Section 3.2, we will show that these factors cannot fully account for the modality gap by simulating an ideal scenario where all of the above factors are controlled for, but the gap still exists at the end of training. The goal of this work is to suggest that the *modality gap* is not caused by trying to align different modalities in a unified representational space, but instead arises as a consequence of the contrastive training that is used to align both modalities. We therefore suggest that the term *contrastive gap* better describes the underlying phenomenon. We define the contrastive gap as the separation between the embeddings produced by two encoders trained using a multimodal contrastive loss (Equation 1). This gap occurs because the embeddings from the two encoders arrange as two disjoint clusters within the shared representational space.

|  | At initialization | After 25 epochs |
|---|---|---|
| Centroid distance | 0.01 | 0.34 |
| Linear separability acc. | 0.52 | 1.00 |
| Contrastive loss | 3.42 | 0.00 |

Table 1: **Modality gap persists even when all factors are controlled for:** Modality gap metrics and CLIP loss values before and after training CLIP from scratch in ideal dataset conditions. **At initialization:** Distance between image and text centroids is almost zero and the embeddings are *not* linearly separable, meaning that there is no modality gap. **After training:** Centroid distance increases slightly, but the text and image embeddings are *perfectly* linearly separable. Thus, the modality gap is *created* by the contrastive loss.

## 3.1 MEASURING THE GAP

To show that we have closed the gap between the embeddings of the two encoders, we must first find a way to quantify the gap. We introduce the following two metrics to measure the size and severity of the gap:

**Distance between modality centroids** (from Liang et al. (2022)) Given $N$ images and $N$ captions, we denote the centroid of the image embeddings as $C^I = 1/N \sum_{j=1}^{N} E_j^I$, and similarly for the centroid of the text embeddings. We compute the distance between centroids as $\|C^I - C^T\|^2$: this distance can vary from 0 to 2, with 0 distance meaning there is no gap between the embeddings.

**Linear Separability** (from Shi et al. (2023)) is the percentage of image and text embeddings that can be distinguished by a linear classifier operating in CLIP space. We used $80\%$ of the dataset to train a linear model to classify CLIP embeddings as originating from either "image" or "text" input. We then tested the performance of the classifier on the remaining $20\%$ of the dataset and reported the accuracy. If a set of embeddings are $100\%$ linearly separable, this means that the space occupied by each modality is completely disjoint, with a clear gap between the embeddings. Conversely, $50\%$ linear separability means that the image and text embeddings are overlapping in CLIP space; i.e. there is no gap between the embeddings.

## 3.2 THE MODALITY GAP PERSISTS EVEN WHEN ALL FACTORS ARE CONTROLLED FOR

We now systematically remove the factors commonly known to contribute to the modality gap. We started with the default CLIP architecture and used the MSCOCO (Lin et al., 2014) dataset. We created an idealized scenario where:

1. *There is only one modality.* We replaced the text encoder in CLIP with another copy of the image encoder and trained the model on pairs of images instead of text-image pairs. Thus, for this experiment, the CLIP encoders are identical image encoders with different random initializations.
2. *Embeddings from the two image encoders occupy the same cone at initialization.* In our setup, after we initialize the two image encoders, we computed a fixed transformation matrix that translates the embeddings of the second image encoder to overlap with those of the first image encoder (following Liang et al. (2022)). Thus, the second encoder's embeddings are translated to occupy the same narrow cone as the first encoder's embeddings at initialization. This way, there is no modality gap at initialization.
3. *There are no mismatched pairs.* The positive pairs in our constructed dataset are actually identical images. This eliminated the possibility that there would be mismatched pairs in the dataset.

We ran this experiment on the full MS COCO training set. We used a batch size of 64, and the default CLIP dimensionality of 512D (i.e. Image embeddings, $E^I \in \mathbb{R}^{512}$). We trained the model for 25 epochs and noted the final training loss value as close to zero.

We observe in Table 1 that even when the dataset is idealized and almost trivial to optimize on, and the model is initialized without a gap, after training to zero loss the CLIP embeddings are perfectly

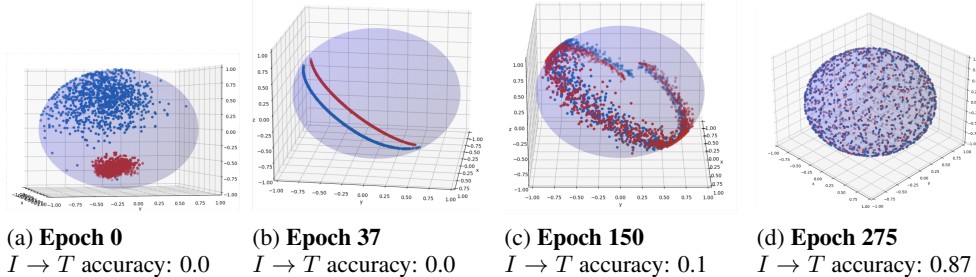

(a) **Epoch 0**
$I \rightarrow T$ accuracy: 0.0

(b) **Epoch 37**
$I \rightarrow T$ accuracy: 0.0

(c) **Epoch 150**
$I \rightarrow T$ accuracy: 0.1

(d) **Epoch 275**
$I \rightarrow T$ accuracy: 0.87

Figure 1: Visualizing the training stages of 3D CLIP on 1000 image-text pairs from MS COCO. Red points are image embeddings, and blue points are text embeddings. $I \rightarrow T$ accuracy represents the text retrieval accuracies. The embeddings are initialized to reside in separate cones due to the cone effect. They then form *arcs*, eventually merge together as rings, and spread out to fill the sphere.

linearly separable. Thus, there is a contrastive gap that is *created* as a byproduct of the contrastive loss, even when the two encoders learn to align the same modality, with overlapping initializations and no mismatched pairs.

### 3.3 VISUALIZING THE GAP IN 3D CLIP

We further investigate the persistence of the gap in CLIP space by training CLIP in 3-dimensional space. Surprisingly, in this lower-dimensional setting, we found that the contrastive gap can be closed even in a non-idealized setting (similar to typical CLIP training). To illustrate this, we visualized the training process using 1,000 randomly sampled image-text pairs from the MS COCO dataset in Figure 1. This finding suggests that closing the contrastive gap may be easier in lower-dimensional spaces vs. in high-dimensional spaces.

We report the text-retrieval accuracies in Figure 1 which shows how accurately we can retrieve an image's caption using the image's embedding and comparing it against all the caption embeddings in the dataset. We see that points on the 3D sphere are best aligned when they are evenly distributed on the sphere (as evidenced by the very high text-retrieval accuracy when Figure 1d). Therefore, we speculate that it is desirable to close the contrastive gap *and* to distribute the embeddings more uniformly on the unit sphere in $\mathbb{R}^d$.

### 3.4 WHY DOES THE CONTRASTIVE LOSS INDUCE A GAP?

In Section 3.2, we observed that the contrastive gap persists even when training CLIP on idealized conditions. We speculate that this is because of the small value of the learned temperature parameter $\tau = 0.01$ in CLIP. As shown by Wang & Liu (2021b), when the temperature approaches zero, the contrastive loss becomes a triplet loss with a margin of zero. This means that the loss only focuses on the nearest negative sample within the batch: Once the similarity between a positive pair is greater than that between the positive and the nearest negative sample, the loss stops pushing the positive pair closer together. Wang & Liu (2021b) uses the following approximation to illustrate this point:

$$\lim_{\tau \to 0^+} -\log \left[ \frac{\exp(s_{i,i}/\tau)}{\sum_{k=1}^{M} \exp(s_{(i,k)}/\tau)} \right]$$
$$= \lim_{\tau \to 0^+} \frac{1}{\tau} \max[s_{\max} - s_{i,i}, 0] \tag{2}$$

Where $s_{(i,i)}$ is the similarity between the $i$th sample and its positive pair, $s_{(i,k)}$ is the similarity between the $i$th sample and the $k$th sample (forming a negative pair), and $s_{\max}$ is the similarity between the $i$th sample and the nearest negative pair to the $i$th sample (i.e, the maximum of all the negative similarities). While a reasonable idea would be to increase the temperature to close the gap between embeddings, this requires manual hand-tuning for each downstream task (Oh et al., 2023; Al-Jaff, 2023), and introduces a trade-off between uniformity and alignment (Wang & Liu, 2021a).

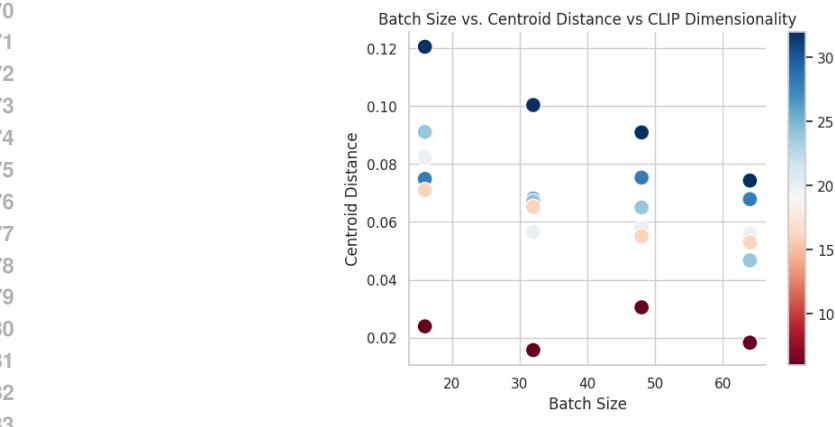

Figure 2: Interplay between batch size, CLIP dimensionality, and size of the contrastive gap. Points trend downwards towards the right (as batch size increases) and are more red (lower CLIP dimensionalities) towards the bottom. This indicates that large batch sizes and smaller CLIP dimensionalities lead to smaller contrastive gaps.

Further, Udandarao (2022) shows that increasing temperature to close the gap between embeddings could degrade the quality of the representational space and hurt downstream task performance.

### 3.5 INTERPLAY BETWEEN DIMENSIONALITY AND BATCH SIZE IN CLIP

Referring to Equation 2, one way to reduce the gap would be to increase $s_{\max}$ which would push negative samples closer to the positive pair. This, in turn, would force the loss to increase $s_{(i,i)}$, potentially closing the gap. Following this reasoning, we can analyze how dimensionality and batch size might influence the contrastive gap by considering their impact on $s_{\max}$:

1. **Increasing batch size** adds more negative samples to the mini-batch, which raises the chance of having more similar negative samples. Let $s_{\max}^m$ represent the maximum similarity to a negative pair for a batch size of $m$. If we increase the batch size to $m + 1$, we have $s_{\max}^m \leq s_{\max}^{m+1}$, as the additional negative sample can only increase or maintain the maximum similarity. As a result, increasing batch size is likely to increase $s_{\max}$ and thereby *reduce* the contrastive gap.
2. **Increasing dimensionality of CLIP** expands the space in which the embeddings are positioned, represented by a unit hypersphere. As dimensionality increases, the distance between randomly selected points on the unit sphere converges to $\sqrt{2}$, and most points start to get far apart (Blum et al., 2020). This means that in higher dimensions, $s_{\max}$ tends to decrease. Consequently, increasing CLIP dimensionality is likely to *increase* the contrastive gap.

We empirically demonstrate in Figure 2 how the interplay between CLIP dimensionality and batch size affects the contrastive gap. Our findings confirm that smaller batch sizes and higher dimensionalities are associated with a larger contrastive gap. This helps explain why the contrastive gap is so common in multi-modal contrastive models: They are trained with relatively small batch sizes in very high-dimensional spaces.

Wang & Isola (2020) show that the uni-modal contrastive loss, in the limit of infinite batch size, asymptotically optimizes for two properties: *uniformity* and *alignment*. In the next section, we adapt these properties to multi-modal contrastive learning and incorporate them into our loss function to reduce the contrastive gap without needing to increase batch size.

## 4 MULTIMODAL UNIFORMITY AND ALIGNMENT

We now introduce the concepts of uniformity and alignment in the representational space. *Uniformity* refers to the property of the embeddings being uniformly distributed throughout the contrastive

latent space. *Alignment* refers to the positive pairs being close together (aligned) in the latent space. To study the effects of closing the contrastive gap, we adapt the uniformity and alignment properties from Wang & Isola (2020) to the multi-modal contrastive space as follows:

$$\text{Uniformity for Image space}: L_{\text{Uniform}}^I = \log\left(\frac{1}{N}\sum_{j=1}^{N}\sum_{k=1}^{N}\exp\left(-2\|E_j^I - E_k^I\|^2\right)\right) \tag{3}$$

We define the uniformity for text space ($L_{\text{Uniform}}^T$) similarly. Finally, the total $L_{\text{Uniform}}$ term is:

$$L_{\text{Uniform}} = \frac{1}{2}(L_{\text{Uniform}}^T + L_{\text{Uniform}}^I) \tag{4}$$

$L_{\text{Uniform}}^I$ and $L_{\text{Uniform}}^T$ each encourage the uniformity *within* the image and text embeddings respectively. i.e., $L_{\text{Uniform}}$ only encourages *intra-modality* uniformity. The original multi-modal contrastive loss (Equation 1) does *not* have any such term that constrains embeddings within each modality to be far apart. Instead, the denominators in Equation 1 only push negative text samples away from the positive image sample and vice versa.

To enforce a stronger constraint on the uniformity *between* negative image and text samples, we also introduce a *cross-modality uniformity* term:

$$\text{Cross-modality uniformity}: L_{\text{XUniform}} = \log\left(\frac{1}{N}\sum_{j=1}^{N}\sum_{k=1,k\neq j}^{N}\exp\left(-2\|E_j^I - E_k^T\|^2\right)\right) \tag{5}$$

Finally, to better align positive image-text pairs in CLIP space, we adapt the alignment term to the multi-modal setting:

$$L_{\text{Align}} = \frac{1}{N}\sum_{j=1}^{N}(\|E_j^I - E_j^T\|^2) \tag{6}$$

Uniformity and alignment properties have been shown to be desirable properties in the uni-modal contrastive space (Wang & Isola, 2020). We validate the desirability of these two properties in the multi-modal setting. In prior works evaluating the multi-modal contrastive representational space using alignment and uniformity properties, Goel et al. (2022) and Oh et al. (2023) used cross-modality uniformity to measure uniformity in the latent space (after fine-tuning with their proposed $m^2$-mix loss). Meanwhile, Al-Jaff (2023) explored explicitly training multi-modal models using $L_{\text{Uniform}}$ (i.e., only the intra-modality uniformity term) and $L_{\text{Align}}$. In our work, we combine both the intra-modality and cross-modality uniformity terms to encourage more uniformity in the latent space and reduce the size of the contrastive gap.

## 5 EXPERIMENTS

We studied the effects of reducing the contrastive gap in CLIP space by optimizing for uniformity and alignment of the latent space. We fine-tuned a pre-trained CLIP model by adding the $L_{\text{Uniform}}$, $L_{\text{XUniform}}$, and $L_{\text{Align}}$ terms to the original CLIP loss ($L_{\text{CLIP}}$, Equation 1). Additionally, we compared to the $m^2$-mix loss from Oh et al. (2023). The $m^2$-mix loss requires that we generate synthetic hard negatives by interpolating (*mixing*) between the image and text embeddings. These newly generated synthetic samples are used as in place of the original in-batch negatives. For each of the experiments, we also compared the performance of fine-tuned CLIP with the original pre-trained CLIP model from OpenAI (we termed this as OpenAI-Pretrained).

We demonstrate the effects of fine-tuning pre-trained CLIP on the following losses:

- $L_{\text{CLIP}}$: The default CLIP loss
- $L_{\text{CUA}}$: $L_{\text{CLIP}} + L_{\text{Uniform}} + L_{\text{Align}}$

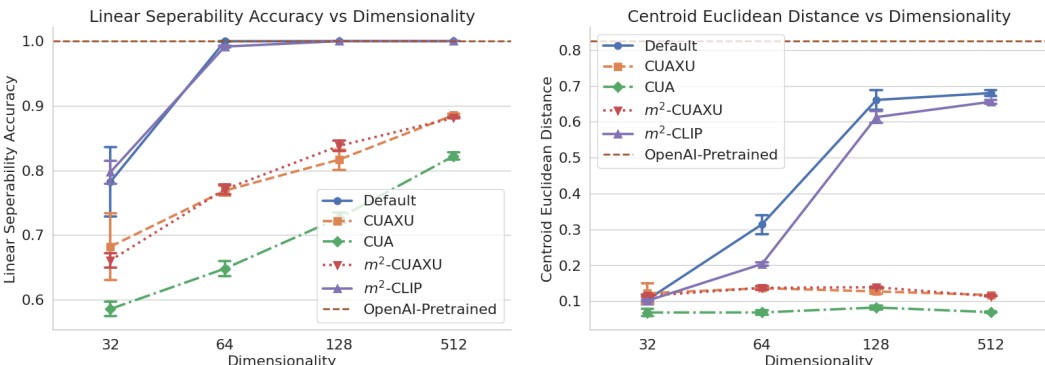

(a) Linear Separability Accuracy vs. CLIP dimensionality.

(b) Distance between image and text centroids vs. CLIP dimensionality.

Figure 3: Contrastive gap metrics after fine-tuning with the different losses for different CLIP dimensionalities.

- $L_{\text{CUAXU}}$: $L_{\text{CLIP}} + L_{\text{Uniform}} + L_{\text{Align}} + L_{\text{XUniform}}$
- $L_{m^2\text{-CLIP}}$: $L_{m^2\text{-mix}} + L_{\text{CLIP}}$
- $L_{m^2\text{-CUAXU}}$: $L_{m^2\text{-mix}} + L_{\text{CLIP}} + L_{\text{Uniform}} + L_{\text{Align}} + L_{\text{XUniform}}$

**Hyperparameters Overview** For our experiments, we fine-tuned the CLIP model made available by Radford et al. (2021). We studied the effects of fine-tuning over various sizes of CLIP space ($\mathbb{R}^d$, $d \in [32, 64, 128, 512]$). We adjusted the dimensionality of CLIP by randomly re-initializing the final linear projection layer. For all our experiments, we fixed the temperature ($\tau$) parameter to 0.01, as $\tau$ converges to this value after CLIP pre-training. We fine-tuned on the MS COCO (Lin et al., 2014) dataset for 9 epochs. Additionally, to provide a stronger proof-of-concept of our proposed methods, we fine-tuned two 128-dimensional models using $L_{\text{CLIP}}$ and $L_{\text{CUAXU}}$ on the much larger Conceptual Captions (Sharma et al., 2018) dataset and report the values of all metrics in Figure 10 in Appendix A.5. We list all hyperparameter settings in the Appendix B.3.

## 5.1 EFFECTS OF THE NEW LOSSES ON THE SIZE OF THE CONTRASTIVE GAP

First we explore the contrastive gap metrics after fine-tuning on the new losses. We compute the gap metrics on the MS COCO validation dataset (5k image-caption pairs). We use the first of the five captions available per image in MS COCO. Figure 3a shows the linear separability accuracy and Figure 3b shows the distance between centroids vs. CLIP dimensionality for each of the different losses. Firstly, we observe that the contrastive gap is larger in OpenAI's pretrained CLIP than in all of our fine-tuned baselines, as evidenced by the larger distance between image and text centroids in Figure 3b. Next, we observe that $L_{\text{CUA}}$, $L_{\text{CUAXU}}$, and $L_{m^2\text{-CUAXU}}$ have lower measures of both the metrics of the contrastive gap when compared to $L_{\text{CLIP}}$ and OpenAI's pretrained CLIP. This indicates that the size of the gap is much smaller with uniformity and alignment terms included. The differences in the size of the contrastive gap are more pronounced in higher CLIP dimensionalities. These results support our claim that increasing uniformity and alignment in CLIP space reduces the size of the contrastive gap. Further, it appears that directly optimizing for uniformity and alignment may lead to smaller contrastive gaps, compared to learning with synthetic hard negatives, as done in Oh et al. (2023).

## 5.2 EFFECTS OF THE NEW LOSSES ON IMAGE-TEXT RETRIEVAL

Next, we evaluated the quality of the representational space after reducing the contrastive gap. We measured the image-text retrieval performance on ECCV-Captions dataset. The MS COCO dataset has many missing correspondences between images and captions: While one caption has only one matching image, that same caption could be a equally plausible description for another image in the dataset, and vice versa. The presence of these *false negatives* lead to a misrepresentation of the image-text retrieval performance if measured directly on MS COCO: We give examples in Figure 9

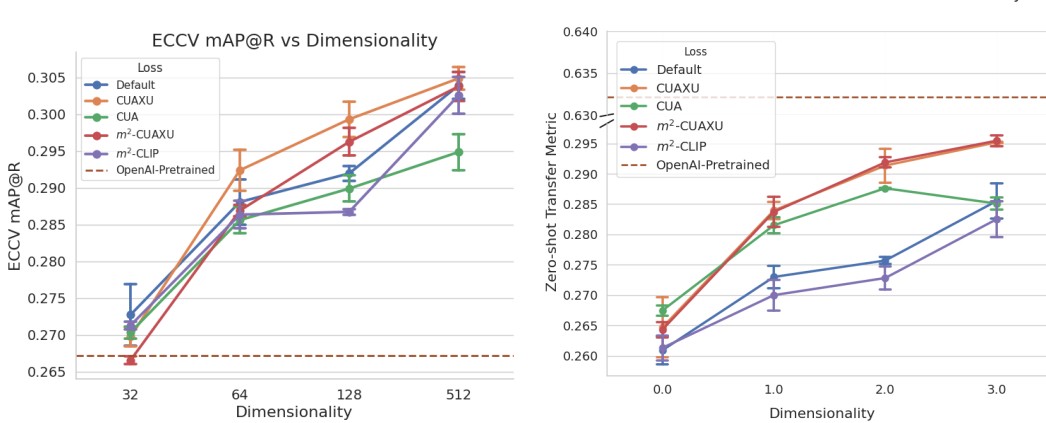

Figure 4: mAP@R for fine-tuned CLIP on different losses on the ECCV-Captions dataset.

Figure 5: Average zero-shot classification accuracies for fine-tuned CLIP on the different losses.

in Appendix A.3. We also show the Top5 image and text retrieval accuracies as measured directly on the MS COCO dataset in Figure 8 in Appendix A.3.

ECCV-Captions extends the MS COCO dataset by adding the missing image-caption correspondences using human and machine annotators. We report the mAP@R metric for average image-text recalls in Figure 4. mAP@R is more aligned to humans than TopK Recall, and is therefore the metric recommended by Chun et al. (2022). Figure 4 shows that $L_{\text{CUAXU}}$ slightly outperforms other baselines. However, we find that $L_{\text{CUA}}$ gives worse retrieval performance than $L_{\text{CLIP}}$. The reduced contrastive gap in the representational space (in the case of $L_{\text{CUAXU}}$) might lead to more accurate image-text correspondences. However, it is important to optimize for cross-modality uniformity, and therefore drive negative image-text pairs further apart, to perform well in the image-text retrieval task (as evidenced by the poor performance of $L_{\text{CUA}}$). Additionally, we see that fine-tuning on MS COCO generally improves performance of all baselines over OpenAI's pretrained CLIP.

## 5.3 ZERO-SHOT TRANSFER

We evaluate our fine-tuned CLIP models on 13 standard image-classification datasets (listed in the Appendix A.4). We also describe the details of the evaluation strategy in A.4. Figure 5 shows the average zero-shot accuracies across all the datasets for each of the losses and dimensionalities. CLIP losses with alignment/uniformity consistently outperform the default CLIP loss, and XUniform adds additional benefit. Thus, we argue that representational spaces with smaller contrastive gaps (as learned by models fine-tuned with added $L_{\text{CUA}}$ and $L_{\text{CUAXU}}$) appear to correlate with higher performance for the zero-shot transfer task. We reason that a smaller contrastive gap likely leads to a better representation of "concepts" within images, which benefits zero-shot classification and average image-text retrieval performances.

We note that fine-tuning CLIP on any of the losses on the MS COCO dataset reduces the average zero-shot transfer performance compared to OpenAI's pretrained CLIP on the datasets we tested. MS COCO is a much smaller dataset than the one OpenAI's CLIP was pretrained on, covering fewer visual concepts. Fine-tuning CLIP on MS COCO shifts the model's representational space to fit MS COCO's distribution. This can lead to a loss of generalization to the datasets we tested. Despite this limitation, our experiments show that $L_{\text{CUAXU}}$ outperforms $L_{\text{CLIP}}$ on average zero-shot image classification when fine-tuned on the same MS COCO dataset. We hypothesize that $L_{\text{CUAXU}}$'s ability to reduce the contrastive gap by encouraging better uniformity and alignment in the representational space leads to more generalization capability, even when fine-tuning on smaller datasets.

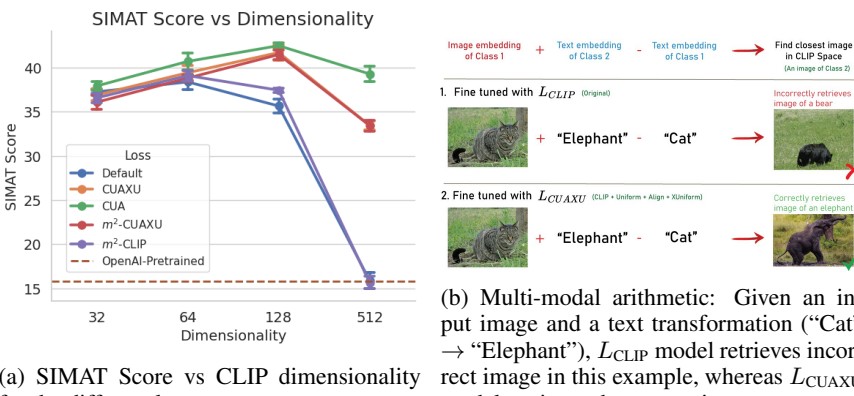

(a) SIMAT Score vs CLIP dimensionality for the different losses.

(b) Multi-modal arithmetic: Given an input image and a text transformation ("Cat" → "Elephant"), $L_{\text{CLIP}}$ model retrieves incorrect image in this example, whereas $L_{\text{CUAXU}}$ model retrieves the correct image.

Figure 6: Multi-modal arithmetic evaluation of models fine-tuned with the different losses.

## 5.4 MULTIMODAL ARITHMETIC

A high-quality multi-modal representational space should have consistent structural representations across the learned modalities. We used SIMAT (Semantic IMage Transformation) (Couairon et al., 2022) to evaluate such relationship consistencies between CLIP image and text embeddings. SIMAT computes a new image representation after transforming it with *text delta vectors* and retrieves the closest image to the transformed embedding. (i.e., $E^I_{\text{target}} = E^I_{\text{input}} + \lambda(E^T_{\text{target}} - E^T_{\text{input}})$).

Figure 6a shows SIMAT scores across the different loss functions. $L_{\text{CUA}}$ and $L_{\text{CUAXU}}$ result in increased SIMAT scores, indicating that the added uniformity and alignment terms result in a more interpretable mapping between modalities. We also observe that all fine-tuned baselines (Except $L_{\text{CLIP}}$ at 512D) perform much better in the multimodal arithmetic task than OpenAI's pretrained CLIP. Notably, the difference in SIMAT scores between $L_{\text{CLIP}}$ and the modified losses becomes more pronounced as CLIP dimensionality increases. We see a similar trend in Figures 3a and 3b, where $L_{\text{CUA}}$ and $L_{\text{CUAXU}}$ lead to representational spaces with significantly lower contrastive gap than in $L_{\text{CLIP}}$, especially in higher dimensionalities. Figure 6b illustrates the multi-modal arithmetic task.

From these empirical findings, we conclude that having a smaller contrastive gap is well correlated with higher performance on the multi-modal arithmetic task. Our results suggest that closing the contrastive gap by fine-tuning with added uniformity/alignment terms could benefit applications that rely on the geometric structure and consistent arithmetic properties in the latent space.

## 6 CONCLUSIONS

We studied the representations of multi-modal contrastive learning and the contrastive gap. Our analysis showed that controlling for the reasons thought to cause the gap does *not* close it. Thus, this is *not a modality gap*. We instead proposed the term *contrastive gap* to describe this phenomenon.

We investigated the relationship between dimensionality and batch size, concluding that the contrastive gap is exacerbated by small batch sizes in high dimensional CLIP space. To reduce the gap, we proposed adding uniformity and alignment terms to the CLIP loss. Our results demonstrated that directly optimizing for uniformity and alignment in the latent space significantly reduces the gap. Moreover, we found that closing the gap through fine-tuning with $L_{\text{CUAXU}}$ improved average image-text retrieval and zero-shot image classification performances, suggesting that reducing the contrastive gap could lead CLIP to develop a more accurate understanding of concepts in images. Further, the new losses also improved multi-modal arithmetic performance, suggesting that the new representational space could have more consistent geometric structure.

While our experiments focused on fine-tuning on MS COCO, we hope to expand our scope and train CLIP models from scratch to better understand the impact of uniformity and alignment at scale.

# 7 REPRODUCIBILITY STATEMENT

We now highlight our efforts to ensure the reproducibility of our results. We included the code we used to produce our results in the supplemental materials section of this submission. Further, we specify the hyperparameters used in our experiments in Section 5 and in the Appendix B.3. We also specify the compute hardware we used for our experiments in the Appendix B.2. Moreover, we use publicly available model architectures, and we specify the sources of their implementations in the text whenever applicable. Finally, we use freely available datasets for all our experiments, and specify the dataset details in Section 5 and in the Appendix (A.4, B.1)

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

# A APPENDIX

## A.1 MOTIVATION BEHIND TRAINING CLIP IN 3D

We train CLIP in 3D in Section 3.3 to better visualize the persistence of the gap on the 3D hypersphere. Our study builds on the work in Shi et al. (2023), where the authors also studied the behaviour of CLIP loss by visualizing training on a 3D sphere. The authors devised this proof of concept experiment to show that the CLIP loss induces a gap even in low dimensional space without external factors coming from the dataset or neural network architecture. They randomly generated 1,000 points on the 3D sphere and trained with a batch size of 10, learning rate of 0.01, using the SGD optimization algorithm. The parameter space for the SGD optimization algorithm was the euclidean coordinate space of the generated points. This allowed the algorithm to directly optimize the positions of the points to reduce the CLIP loss.

Their experiment highlights that the modality gap can arise even in low-dimensional spaces and without any external factors from the dataset or neural network architecture, but can eventually be closed after a large number of training steps.

Shi et al. (2023), we studied the behaviour of CLIP loss when we reduce the CLIP dimensionality from 512D to 3D. We extended the proof-of-concept experiment done by Shi et al. (2023) by

- **Using Real Data**: We sampled 1,000 image-text pairs from the MS-COCO dataset rather than using synthetic points. This way, we can better capture the nuances of a real-world data distribution.

- **Optimizing with CLIP architecture**: In the original experiment by Shi et al. (2023), the authors bypassed the neural network entirely by directly optimizing the positions of points in a 3D Euclidean space. However, in our modified experiment, we utilized the CLIP model to project image and text representations into a 3D space (by adjusting the output projection layer from 512D to 3D). By doing so, we could examine how the CLIP model and its loss function behave together, allowing us to assess the impact of both the neural network and the loss function in a more realistic setting.

We visualize the results of our 3D experiment in Section 3.3 of the main text.

## A.2 MEASURING DISTRIBUTION OF EMBEDDINGS ON THE CLIP HYPERSPHERE

We showed in Section 5.1 of the main text the effects of the various new losses on the size of the contrastive gap between image and text embeddings. Here, we evaluate the distribution of embeddings across latent space dimensions by applying Principal Component Analysis (PCA) and analyzing the explained variance ratios (Holland, 2008)

PCA is a dimensionality reduction technique that summarizes the data into a smaller set of principal components (PCs). The explained variance ratio indicates how much of the original data's variance is captured, or "explained", by each PC. By examining the cumulative PCA explained variance curve, we can assess how well the embeddings are spread across the dimensions of the unit hypersphere in CLIP space. Ideally, if the embeddings are uniformly distributed across all dimensions—indicating high uniformity—the cumulative PCA explained variance curve will form a straight line. Figure 7 shows the cumulative PCA explained variances for the three different losses for CLIP space in $\mathbb{R}, d \in [32, 64, 128]$.

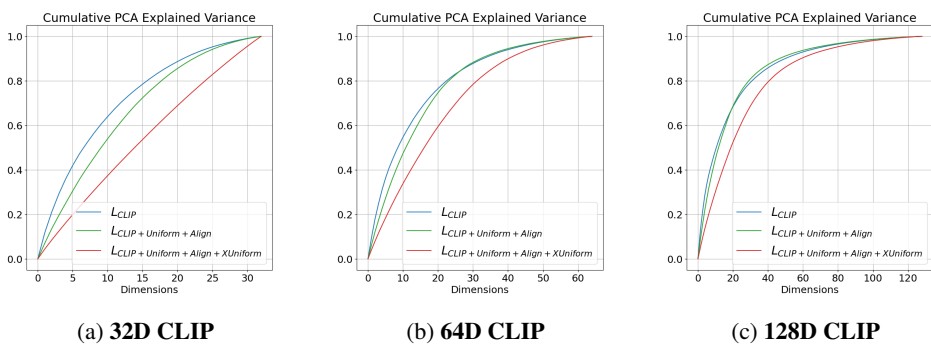

(a) **32D CLIP**      (b) **64D CLIP**      (c) **128D CLIP**

Figure 7: PCA explained variances for each CLIP dimensionality after fine-tuning

From the figure, we see that $L_{\text{CUAXU}}$ has the lowest cumulative variance across all the principle components for all the three dimensionalities of CLIP space tested. This indicates that the the uniformity terms help to distribute the embeddings along more dimensions in the contrastive latent space in $\mathbb{R}^d$, across a wide range of $d$'s

## A.3 IMAGE-TEXT RETRIEVAL PERFORMANCES

Figure 8 shows the image and text retrieval accuracies for MS COCO validation set after fine-tuning CLIP on the different losses. As described in the main text in Section 5.2, MS COCO dataset has many false negatives, which may misrepresent the retrieval performance of the models. We explain this issue in more detail here.

We speculate that smaller contrastive gap leads to better representation of "concepts" within images. However, in image-retrieval and text-retrieval tasks, the model needs to precisely match the caption to the exact ground truth image. Even if the model retrieves an image with the correct concepts, it will be considered incorrect if it does not match the exact ground truth image. Figure 9 demonstrates this, illustrating two examples where models fine-tuned with both $L_{\text{CLIP}}$ and $L_{\text{CUAXU}}$ retrieves the "wrong" caption for the input image. We observe that even though the retrieved captions do not match the ground truth captions exactly, they are not necessarily wrong, as they encapsulate the correct concepts that are present in the images.

The ECCV-Caption dataset accounts for these false negatives. Therefore, we use ECCV-Caption to measure image-text retrieval performance of our fine-tuned models in Section 5.2.

## A.4 ZERO SHOT TRANSFER DATASETS

We evaluate our fine-tuned CLIP models on the standard image-classification datasets outlined in Table 2. To test the zero-shot transfer capabilities of our CLIP models, we adopt the evaluation strategy of Goel et al. (2022), which is also recommended by Radford et al. (2021): We generate

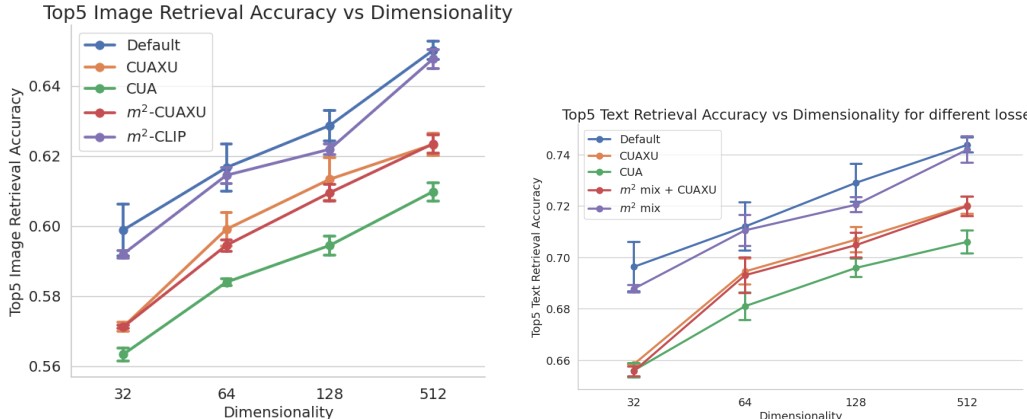

(a) Top5 Image Retrieval Accuracy ($T \to I$) vs CLIP dimensionality.

(b) Top5 Image Retrieval Accuracy ($I \to T$) vs CLIP dimensionality.

Figure 8: Image and text retrieval accuracies for different CLIP dimensionalities.

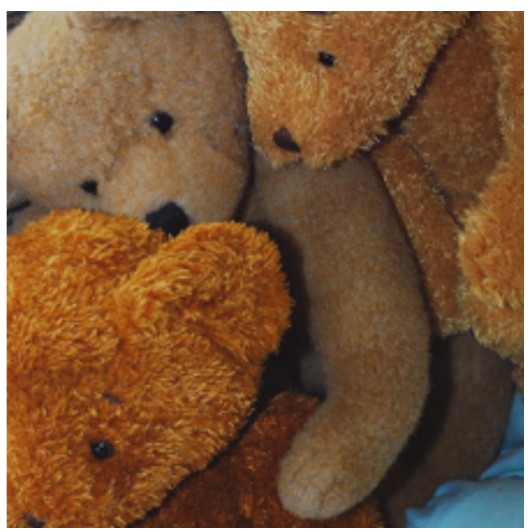 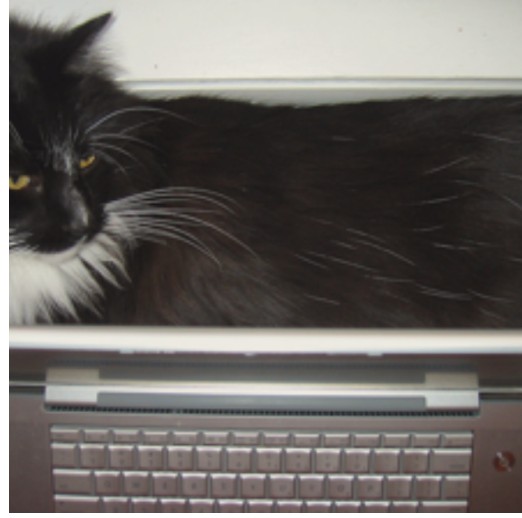

(a) **Ground Truth Caption:** Three stuffed animals are sitting on a bed.
**Retrieved Caption using $L_{\mathrm{CLIP}}$ :** This is a picture of four stuffed animals.
**Retrieved Caption using $L_{\mathrm{CUAXU}}$:** Two teddy bears are seated with the other stuffed animals.

(b) **Ground Truth Caption:** A black fluffy cat sitting on top of a computer keyboard.
**Retrieved Caption using $L_{\mathrm{CLIP}}$ :** A cat looking up at the television that has something interesting on it.
**Retrieved Caption using $L_{\mathrm{CUAXU}}$:** Gray and white cat sitting next to open laptop.

Figure 9: Illustrating two instances where models fine-tuned with both $L_{\mathrm{CLIP}}$ and $L_{\mathrm{CUAXU}}$ make a mistake in retrieving the nearest caption for a given image.

prompts using the class names to form sentences like "a photo of a {class name}", "A sketch of a {class name}", etc. We then pass these sentences through the text encoder to get prompt embeddings. We average all the prompt embeddings to get a class embedding for each class. Finally, to classify an image, we pass the input image through the image encoder and obtain its image embedding. We then determine the predicted class by finding the class embedding closest to the image embedding using cosine similarity.

| Dataset | Classes | Test size | Evaluation metric |
|---|---|---|---|
| CIFAR-10 | 10 | 10,000 | Accuracy |
| CIFAR-100 | 100 | 10,000 | Accuracy |
| SUN397 | 397 | 19,850 | Accuracy |
| Pascal VOC 2007 | 20 | 4,952 | 11-Point mAP |
| Oxford-IIIT Pets | 37 | 3,669 | Mean Per Class |
| Caltech-101 | 102 | 6,085 | Mean Per Class |
| ImageNet | 1000 | 50,000 | Accuracy |
| ImageNet-V2 | 1000 | 10,000 | Accuracy |
| ImageNet-Sketch | 1000 | 50,000 | Accuracy |
| ImageNet-A | 200 | 7,500 | Accuracy |
| ImageNet-R | 200 | 30,000 | Accuracy |
| ImageNet-O | 200 | 2,000 | Accuracy |
| ObjectNet | 113 | 50,000 | Accuracy |

Table 2: Datasets evaluated on to test zero-shot image-classification performance of the different CLIP losses.

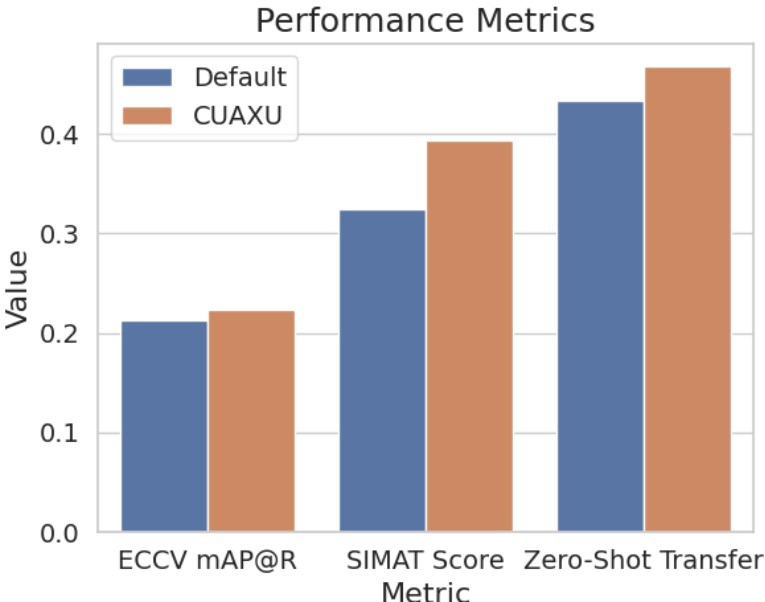

Figure 10: Values of all three evaluation metrics: Image-text retrieval (ECCV mAP@R), zero-shot transfer, and multi-modal arithmetic (SIMAT Score), of 128-dimensional CLIP models fine-tuned with $L_{\text{CLIP}}$ and $L_{\text{CUAXU}}$. The model fine-tuned with $L_{\text{CUAXU}}$ consistently outperforms that fine-tuned with $L_{\text{CLIP}}$.

## A.5 Evaluating on Conceptual Captions Dataset

In order to provide a stronger proof-of-concept of our approach, we evaluated our proposed method on the significantly larger conceptual captions dataset (Sharma et al., 2018). In our experiment, we were able to access about 2.2 million image-caption pairs through image urls specified by the dataset. We fine-tuned two models, one with default CLIP loss, and another with our proposed CUAXU loss. We show in Figure 10 that the CLIP model fine-tuned with CUAXU loss outperforms the model fine-tuned with default CLIP loss across all the three metrics we measured.

| Hyperparameter | Value |
|---|---|
| Image encoder | ViT/B-32 |
| Text Encoder | Transformer (same as in [3]) |
| Embedding dimensions | [32, 64 ,128, 512] |
| Temperature | 0.01 |
| Epochs | 9 |
| Batch size | 64 |
| Learning rate | 1e-6 |
| Adam beta1 | 0.9 |
| Adam beta2 | 0.99 |
| Adam weight decay | 0.1 |
| Scheduler | None |

Table 3: Hyperparameters used for fine-tuning the CLIP models.

## B  EXPERIMENTAL SETUP

### B.1  DATASET

We fine-tune on the MS COCO dataset downloaded from (Lin et al., 2014) [1]. We use the 2017 split, with 118k training images, and 5k validation images. Each image has 5 human-generated captions associated with it. In our experiments, we only take the first caption for each image.

### B.2  COMPUTATIONAL RESOURCES USED

For all our training runs, we use NVIDIA RTX A5000 GPUs, and Intel(R) Xeon(R) Silver 4210 CPUs @ 2.20GHz. One run for fine-tuning CLIP from pre-trained weights on MS COCO for 9 epochs needed about 7GB GPU memory, and needed about 3.5 hours to complete (on a single GPU).

### B.3  MODEL ARCHITECTURE AND HYPERPARAMETERS

We fine-tune the pre-trained CLIP model made available by OpenAI from Huggingface (Wolf et al., 2020)[2]. We list the model hyperparameters that we use below:

## C  LIMITATIONS

In this work, we explore the properties of uniformity and alignment in the multi-modal setting, showing that added uniformity and alignment terms help to reduce the contrastive gap. One limitation of our work is that we show this by fine-tuning CLIP on the MS COCO and Conceptual Captions datasets. Training CLIP from scratch instead of fine-tuning CLIP may help gain more insights into how the contrastive loss emerges during training.

---

[1]https://cocodataset.org
[2]https://huggingface.co/docs/transformers/en/model_doc/clip

