# OpenReview forum: "It's Not a Modality Gap: Characterizing and Addressing the Contrastive Gap"
_ICLR.cc/2025/Conference — Submitted to ICLR 2025_

### Official Review · Reviewer_iaqR · 2024-10-30

**Soundness:** 3
**Presentation:** 3
**Contribution:** 3
**Rating:** 6
**Confidence:** 5

**Summary:**

This paper explores the problem of modality gap in contrastive pre-trained models such as CLIP. This paper shows experimentally that the contrastive loss itself can also introduce gaps during training. This contrastive gap is particularly pronounced when training with small batches in high-dimensional spaces. To address this problem, this paper proposes a method to minimize the contrastive gap by adding uniformity and alignment terms. Experimental results show that this approach can optimize the representation space and achieve better performance in downstream tasks such as zero-shot image classification and multimodal arithmetic.

**Strengths:**

- Simple experiments prove the motivation.
- The paper is clearly written and the method can be well understood.

**Weaknesses:**

- Alignment and Uniformity are properties often mentioned in the CLIP model, and the innovation of Section 4 seems to be limited.

- The experimental results of Figure 4 and Figure 5 are completely different. So is the poor image retrieval result related to the dataset? If we use categories as text for retrieval on datasets such as CIFAR-10, the retrieval conclusion will be similar to the conclusion in Figure 5.

**Questions:**

- In Figure 1, why is the accuracy only 0.1 in the first 150 epochs, and then quickly increases to 0.87? This phenomenon seems a bit abnormal. Can you explain the reason?


- Why does increasing batch size and s_max reduce the contrast gap?

---

> ### Author Response · Authors · 2024-11-22
> **Response to Reviewer iaqR**
>
> We thank the reviewer for their helpful review. We first answer their questions, and then address the identified weaknesses below.
>
> ### Answers to Questions:
>
> **In Figure 1, why is the accuracy only 0.1 in the first 150 epochs, and then quickly increases to 0.87? This phenomenon seems a bit abnormal. Can you explain the reason?**
>
> This phenomenon stems from how the embeddings populate the representational space during training. At 150 epochs, embeddings do not have enough space to properly align the image and text pairs in the 3D sphere. As training progresses, the embeddings spread across the hypersphere, reducing the contrastive gap and improving the model’s ability to correctly align image-text pairs.
> This result supports our motivation for increasing uniformity by spreading the embeddings throughout the sphere to improve accuracy.
>
> **Why does increasing batch size and s_max reduce the contrast gap?**
>
> Increasing s_max brings negative image-text pairs closer together. From Eqn (2) in the paper, CLIP loss drives positive image-text pairs to be more similar such that their similarity is greater than s_max. Thus, higher s_max would force CLIP loss to drive positive image-text pairs closer together, reducing the distance between image cluster and text cluster in embedding space, reducing the contrastive gap.
>
> Increasing batch size increases s_max, because additional negative samples can only increase or maintain the maximum similarity between negative image-text pairs within the batch.
>
> Therefore, increasing batch size increases s_max, which in turn reduces the contrastive gap.
>
> ### Addressing Weaknesses
>
> **Limited innovation of Section 4 (uniformity and alignment properties often mentioned in CLIP models)**
>
> It is correct that uniformity and alignment are often mentioned when talking about CLIP in past works. However, to the best of our knowledge, our work is the first to explicitly *optimize* uniformity and alignment in the multi-modal contrastive setting for images and texts. Whereas, previous works only *measured* these properties without incorporating them into the loss function.
> Furthermore, we combine within-modality and cross-modality uniformity terms into a single loss function, whereas prior works only measure either within-modality or cross-modality uniformity properties in isolation.
>
> **Differing experimental results in Figure 4 and Figure 5**
>
> We speculated in the paper that the new losses likely lead to a better representation of concepts within images. We have now verified this by evaluating our method on ECCV-caption [1], which takes into account the presence of false negatives in MS COCO. We added this result to Figure 4 in the updated paper. We find that our method generally outperforms the default CLIP baseline on the retrieval task on ECCV-caption. This new result supports our discussion in Appendix Section A.3, where we suggest that the lower retrieval performance in MS COCO is due to the presence of false negatives in the dataset.
>
> [1] ECCV Caption: Correcting False Negatives by Collecting Machine-and-Human-verified Image-Caption Associations for MS-COCO, Chun et al., ECCV, 2022

---

> ### Author Response · Authors · 2024-12-02
>
> Dear Reviewer iaqR,
>
> We wanted to kindly follow up regarding your review of our submission. We answered your questions and provided additional results to address your concern of poor image retrieval result by solving the dataset issue (as you pointed out) and evaluating on ECCV-Captions dataset.
>
> Please let us know if you need more clarification or discussions from our side.
>
> We kindly request the reviewer to increase their score if we have addressed all their concerns.

---

### Official Review · Reviewer_pp1S · 2024-11-04

**Soundness:** 2
**Presentation:** 2
**Contribution:** 2
**Rating:** 3
**Confidence:** 4

**Summary:**

This paper investigates the phenomenon known as the "modality gap" in multi-modal contrastive learning models like CLIP. The authors propose renaming it to "contrastive gap," arguing that this gap emerges as a consequence of contrastive training rather than modality differences. They demonstrate that the gap persists even when controlling for previously suspected causes and show that it is exacerbated by small batch sizes in high-dimensional spaces. The paper proposes adding uniformity and alignment terms to the CLIP loss to reduce this gap and evaluates the impact on various downstream tasks.

**Strengths:**

- The paper provides a comprehensive empirical analysis of how dimensionality and batch size affect the contrastive gap, offering insights into why this phenomenon occurs in multi-modal models.
- The experiments are extensive, covering multiple evaluation metrics and downstream tasks, including zero-shot classification, retrieval, and multi-modal arithmetic.
- The proposed solution of adding uniformity and alignment terms is relatively simple to implement and shows some improvements in certain tasks.

**Weaknesses:**

## Major Weaknesses
1. Fundamental Misunderstanding of CLIP:
    - The paper incorrectly attributes CLIP's loss function to SimCLR's NT-Xent loss, when CLIP actually builds upon multi-class N-pair loss[^1]
    - This misunderstanding undermines the paper's theoretical foundation and technical credibility
2. Experimental Validity
    - Absence of crucial baselines: no comparison with untuned CLIP or from-scratch training
    - All fine-tuning methods appear to degrade the original CLIP's performance
    - The generalization from 3D to high-dimensional spaces lacks proper justification
3. Theoretical Framework
    - Misinterpretation of Wang & Isola (2020)[^2], incorrectly claiming uniformity and alignment as "desirable properties"
    - Insufficient justification for renaming "modality gap" to "contrastive gap"
    - Lack of causal analysis between gap reduction and performance improvement

## Minor Weaknesses
1. Experimental Design Choices:
    - Using identical images as positive pairs without proper justification
    - Arbitrary selection of only the first caption from MS COCO's five captions
    - Missing comparison with previous modality gap solutions (e.g., Liang et al. 2022[^3])
2. Result Analysis:
    - Inadequate explanation for contradictory results between retrieval and zero-shot classification tasks
    - Insufficient discussion of task-specific performance variations
    - Limited analysis of the trade-offs introduced by the proposed loss terms

[^1]: Sohn, Kihyuk. "Improved deep metric learning with multi-class n-pair loss objective." Advances in neural information processing systems 29 (2016).
[^2]: Wang, Tongzhou, and Phillip Isola. "Understanding contrastive representation learning through alignment and uniformity on the hypersphere." International conference on machine learning. PMLR, 2020.
[^3]: Liang, Victor Weixin, et al. "Mind the gap: Understanding the modality gap in multi-modal contrastive representation learning." Advances in Neural Information Processing Systems 35 (2022): 17612-17625.

**Questions:**

1. Could you explain why using identical images as positive pairs in your controlled experiment provides meaningful insights about the real-world scenario where modalities are genuinely different?
2. Your experiments show performance degradation in retrieval tasks but improvement in zero-shot classification. How do you reconcile these contradictory results with your claim that reducing the contrastive gap generally improves representation quality?
3. Have you conducted experiments comparing your method with the original CLIP model (without fine-tuning) and with models trained from scratch using your proposed loss functions? This would help establish the true effectiveness of your approach.
4. Given that the original paper showing the modality gap (Liang et al. 2022[^1]) demonstrated that the optimal gap size is task-dependent, how do you justify the claim that uniformly reducing the gap is beneficial?

[^1]: Liang, Victor Weixin, et al. "Mind the gap: Understanding the modality gap in multi-modal contrastive representation learning." Advances in Neural Information Processing Systems 35 (2022): 17612-17625.

---

> ### Author Response · Authors · 2024-11-22
> **Response to Reviewer pp1S**
>
> We thank the reviewer for their helpful review. We first answer their questions, and then address the identified weaknesses below.
>
> ### Answers to Questions
> **1. Why identical images as positive pairs?**
>
> Firstly, using only images and finding that the “so-called” modality gap still exists rules out the modalities being different as a cause for the gap. Secondly, using identical images as positive pairs and still finding that the modality gap persists rules out mismatched pairs as the cause for the gap. This is in contrast to the claims of Liang et al. [3], where existence of mismatched pairs was found to be a contributing factor for the modality gap.
>
> By ruling out these factors as the driving forces behind the modality gap, we offer deeper insights into this phenomenon in the real world even when the modalities are different.
>
> **2. Contradictory results**
>
> We discussed (in Appendix A.3) that the lower retrieval performance could be due to false negatives in MS COCO. We have now verified this by evaluating our method on ECCV-caption [1], which corrects these false negatives. We added this result to Figure 4 in the paper. We find that our method generally outperforms default CLIP on ECCV-caption, suggesting that reducing contrastive gap improves representation quality.
>
> **3. Training from scratch**
>
> While we did not conduct from-scratch experiments, we are currently fine-tuning on a larger dataset to more strongly demonstrate the effectiveness of our approach. We hope to include those results in the paper before the discussion window closes.
>
> **4. Justify that reducing gap is beneficial**
>
> Liang et al. 2022 [3] modified gap size by translating the embeddings of a pre-trained model. Our approach differs fundamentally, because the model learns to reduce the gap through fine-tuning with new loss. We justify our claim that reducing the gap is beneficial by showing that zero-shot classification and multimodal arithmetic performances generally improve with smaller gap sizes across multiple dimensionalities of CLIP space.
>
> ### Addressing Weaknesses
>
> **Major Weaknesses**
>
> **1. Fundamental Misunderstanding of CLIP**
>
> We would like to clarify that the NT-Xent (normalized temperature-scaled cross entropy) term was introduced by the SimCLR paper [5] to include the multi-class N-pair loss [2]. The NT-Xent loss better captures the normalization of vectors and temperature scaling in CLIP loss (Prior work [6] also attributed CLIP loss to NT-Xent loss). While we acknowledge that we should have been more clear on this, we believe this is not a fundamental misunderstanding of CLIP, as the reviewer suggests. We updated the text in the introduction to Section 2.1 to more clearly reflect the basis of the CLIP loss.
>
> **2. Experimental Validity**
>
> We use the 3D example to provide some intuition about the training process of CLIP. We did the rest of our experiments on higher dimensional CLIP spaces. Ex: We conducted our experiment in Section 3.2 on full 512D CLIP.
>
> **3. Theoretical Framework**
>
> Wang & Isola (2020) [4] showed a positive connection of uniformity and alignment with task performances through theoretical analysis and extensive experiments, suggesting that these are indeed desirable properties for representations.
> In Section 3.2, where only one modality is considered, we believe “contrastive gap” is more clear than modality gap, reinforcing that this geometric phenomenon is independent of modality.
>
> **Minor Weaknesses**
>
> **1. Experimental Design Choices:**
>
> We select the first caption out of 5 to keep with the paradigm of training with image-caption pairs where there is only one caption per image
>
> **References**
>
> [1] ECCV Caption: Correcting False Negatives by Collecting Machine-and-Human-verified Image-Caption Associations for MS-COCO, Chun et al., ECCV, 2022
>
> [2] Sohn, Kihyuk. "Improved deep metric learning with multi-class n-pair loss objective." Advances in neural information processing systems 29 (2016).
>
> [3] Liang, Victor Weixin, et al. "Mind the gap: Understanding the modality gap in multi-modal contrastive representation learning." Advances in Neural Information Processing Systems 35 (2022)
>
> [4] Wang, Tongzhou, and Phillip Isola. "Understanding contrastive representation learning through alignment and uniformity on the hypersphere." International conference on machine learning. PMLR, 2020.
>
> [5] T. Chen, S. Kornblith, M. Norouzi, and G. Hinton, “A Simple Framework for Contrastive Learning of Visual Representations,” 2020, arXiv: arXiv:2002.05709.
>
> [6] Peiyang Shi, Michael C. Welle, et al.. Towards understanding the modality gap in CLIP. In ICLR 2023 Workshop on Multimodal Representation Learning: Perks and Pitfalls

---

> > ### Comment · Reviewer_pp1S · 2024-11-26
> >
> > Thank you for your detailed response and for addressing the concerns raised in my review. Your explanations regarding the use of identical images as positive pairs provide valuable insights into the persistence of the contrastive gap, and your additional experiments with the ECCV-caption dataset help clarify the contradictory results observed in retrieval tasks.
> >
> > However, some critical concerns remain unaddressed. The absence of crucial baselines, such as comparisons with the original CLIP model without fine-tuning and from-scratch training using your proposed loss functions, continues to limit the assessment of your method's true effectiveness. While you mention ongoing experiments to address this, the current lack of these results makes it difficult to fully evaluate the impact of your approach. Therefore, I maintain my initial rating but encourage you to include these additional experiments to strengthen your paper.

---

> > > ### Author Response · Authors · 2024-11-30
> > > **Response to Reviewer pp1S**
> > >
> > > We thank the reviewer for their comment.
> > >
> > > In this post, we add the results of running our evaluations on untuned CLIP (OpenAI’s pretrained CLIP) to our plots, as suggested by the reviewer. We added these values (as OpenAI-Pretrained) as dotted lines in Figure 3 (contrastive gap metrics), Figure 4 (average image-text retrieval performance), Figure 5 (average zero-shot transfer performance), and Figure 6(a) (multimodal arithmetic).
> > >
> > > We give an outline of our findings in relation to for each of the evaluation metrics below:
> > >
> > > **Size of contrastive gap**
> > >
> > > - Contrastive gap is larger in OpenAI’s pretrained CLIP compared to models fine-tuned with CLIP loss (distance between centroids higher in Figure 3b than all baselines)
> > >
> > > - Contrastive gap in OpenAI’s pretrained CLIP is much larger than that in models fine-tuned with CUA and CUAXU losses, indicating that our losses effectively reduce the gap after fine-tuning.
> > >
> > > **Image-Text retrieval**
> > >
> > > - Fine-tuning CLIP on any of the losses improves average image-text retrieval performance on ECCV-Captions over OpenAI’s pretrained CLIP. This is because the ECCV-Captions dataset is an extension of the MS COCO test dataset, and we fine-tuned the models on the MS COCO train dataset.
> > >
> > > **Zero-shot transfer**
> > >
> > > - Fine-tuning CLIP on any of the losses on the MS COCO dataset reduces the average zero-shot transfer performance compared to OpenAI’s pretrained CLIP on the datasets we tested.
> > >
> > > - However, the pretrained CLIP model from OpenAI was trained on a vast dataset of 400M image-caption pairs, covering a wide range of concepts. In contrast, MS COCO is a much smaller dataset with 118K images, covering fewer visual concepts. Fine-tuning CLIP on MS COCO shifts the model’s representational space to fit the distribution of the smaller dataset. This can lead to a loss of generalization to the datasets we tested.
> > >
> > > - Despite this limitation of fine-tuning on a small dataset, our experiments show that the CUAXU method outperforms the default CLIP loss when fine-tuned on MS COCO. We hypothesize that CUAXU’s ability to reduce the contrastive gap by encouraging better uniformity and alignment in the shared image-text embedding space leads to more generalization capability, even when fine-tuning on smaller datasets.
> > >
> > > **Multimodal Arithmetic**
> > >
> > > - Fine-tuning CLIP on lower dimensional spaces improves multimodal arithmetic performance over OpenAI’s pretrained CLIP. However, fine-tuning 512-dimensional CLIP with the default loss does not improve multimodal arithmetic performance.
> > >
> > > - At the same dimensionality as OpenAI’s pretrained CLIP (512D), fine-tuning with our proposed losses significantly improves multimodal arithmetic performance over OpenAI’s pretrained CLIP.
> > >
> > > - These findings support our discussion in Section 5.4 in the paper, where we concluded that having a smaller contrastive gap is well correlated with higher performance on the multi-modal arithmetic task.
> > >
> > > Finally, while we do not have results for CLIP trained from scratch, we showed that our CUAXU approach outperforms default CLIP on all 3 measured metrics when fine-tuning CLIP with the much larger Conceptual Captions dataset [1] (in Figure 10, Appendix A.5). This provides a strong proof-of-concept, demonstrating that our proposed method improves the quality of the representations learned over default CLIP.
> > >
> > > We kindly ask the reviewer to reassess their evaluation if these new results address their concerns about our method's true effectiveness.
> > >
> > > [1] Sharma, P., Ding, N., Goodman, S., & Soricut, R. (2018). Conceptual Captions: A Cleaned, Hypernymed, Image Alt-text Dataset For Automatic Image Captioning. Proceedings of ACL.

---

### Official Review · Reviewer_Mgbo · 2024-11-04

**Soundness:** 2
**Presentation:** 3
**Contribution:** 3
**Rating:** 5
**Confidence:** 3

**Summary:**

This paper analyzes the contrastive gap, a phenomenon previously referred to as the modality gap in existing research. The authors make two main observations: (1) they demonstrate that this gap also occurs within the same modality, prompting a shift in terminology from "modality gap" to "contrastive gap," and (2) they show that small batch sizes in high-dimensional spaces are particularly susceptible to the contrastive gap. After characterizing the contrastive gap through the use of uniformity and alignment losses, the authors illustrate that addressing this gap can benefit certain downstream tasks. However, it is worth noting that all experiments are conducted in a small-scale fine-tuning setting, which differs significantly from real-world, large-scale pre-training environments. Additionally, while the authors claim benefits for downstream tasks, these results appear limited to a narrow set of tasks, such as classification.

**Strengths:**

1. The paper is well-written and easy to follow.
2. Although the experiment is conducted on a small scale and with restricted fine-tuning areas, the finding that a gap also arises within the same modality is significant and could impact a broad range of applications.
3. The analysis of the rationale behind how contrastive loss induces a gap is clear and reasonable.

**Weaknesses:**

1. Most experiments are conducted on a small-scale training dataset within a fine-tuning setup, which differs significantly from real-world setting. Additionally, the batch size and dimensionality are also limited to a small-scale setting, raising concerns about the applicability of these results to large-scale contexts. Ideally, additional experiments in a more realistic, large-scale setting would strengthen the findings.
2. The proposed method closely resembles existing work (e.g., Al-jaff, 2023), with the primary difference being the inclusion of cross-modal uniformity. This similarity raises questions about the extent of the methodological contribution.
3. While the authors claim that bridging the contrastive similarity gap benefits representation learning, the results in cross-modal retrieval do not demonstrate significant performance improvements, which is concerning. Although the authors argue that their method captures meaningful concepts, the discussions and experiments provided are insufficient to convincingly support this claim.

The examples in Figure 9 appear to be cases of false negatives, a well-known issue in MS-COCO. Evaluating the proposed method on ECCV-caption[1], which addresses false negatives and uses a more robust metric, such as MaP, could provide a stronger demonstration of its effectiveness.

[1] ECCV Caption: Correcting False Negatives by Collecting Machine-and-Human-verified Image-Caption Associations for MS-COCO, Chun et al., ECCV, 2022

**Questions:**

Refer to the weakness section

---

> ### Author Response · Authors · 2024-11-22
> **Response to Reviewer Mgbo**
>
> We thank the reviewer for their helpful review. We address the identified weaknesses below.
>
> **Experiments too small-scale:**
>
> We are currently running experiments on a larger dataset, and expect to add these results to the paper before the author-reviewer discussion window closes. This will help validate the applicability of our work to larger datasets.
>
>
> Further, while we empirically demonstrate the interplay between batch size and dimensionality in a relatively small scale setting, we also theoretically analyze how batch size and dimensionality might affect the contrastive gap in Section 3.5, making our analysis applicable to large-scale contexts.
>
> **Method closely resembles existing work:**
>
> As far as we know, we are the first to show that the modality (“contrastive”) gap phenomenon persists even in the presence of overlapping encoders, without mismatched pairs, and when both modalities are images.
>
> Further, we study the effects of our losses on image-text datasets, unlike in Al-jaff (2023), where the study is done on Multimodal Handwritten Digits (MHD) and the Single-Cell Data Integration Challenge (CITE) datasets. Finally, we show that the inclusion of cross-modality uniformity is significant as:
> -  It improves the average zero-shot transfer performance over $\text{L}_{\text{CUA}}$, and,
> -  Improves the distribution of embeddings in the representational space over $\text{L}_{\text{CUA}}$ (Figure 7 in Appendix A.2),
>
>
> **No significant improvement in cross-modal retrieval**
>
> We thank the reviewer for their suggestion to evaluate our method on ECCV-caption[1]. We added the result of their suggested evaluation to Figure 4 in the paper. We find that our method generally outperforms the default CLIP baseline on the retrieval task on ECCV-caption.
>
> [1] ECCV Caption: Correcting False Negatives by Collecting Machine-and-Human-verified Image-Caption Associations for MS-COCO, Chun et al., ECCV, 2022

---

> > ### Comment · Reviewer_Mgbo · 2024-11-26
> >
> > Thank you for the detailed response. I’ve carefully reviewed the authors' comments.
> >
> >
> > First of all, it’s great to see the ECCV caption results—CUAXU's performance appears to have improved. However, I have one concern: the performance gap seems to diminish significantly when the batch size is either relatively small (32) or large (512). Could you clarify the reason for this?
> >
> > Additionally, I am still awaiting the large-scale results.
> >
> > I believe the claims made in the paper have the potential to be strong, but I remain concerned that the current experiments may not be sufficient to fully support them. I will finalize my recommendation after reviewing the complete results.
> >
> > If possible, I suggest highlighting the revised sections of the paper in a different color. It’s currently challenging to identify which parts have been updated.

---

> > > ### Author Response · Authors · 2024-11-30
> > > **Response to Reviewer Mgbo**
> > >
> > > Thank you for the response.
> > >
> > > **Regarding ECCV Caption Results**
> > >
> > > We would like to clarify that the x axis in Figure 4 (ECCV-Captions result) is the CLIP dimensionality, and not the batch size. To address the reviewer's concern, we hypothesize that the small performance difference for image-text retrieval at 32 and 512 dimensional CLIPs could be for differing reasons:
> > > - **Small performance difference at 32D**: We suggest this is because the difference between the contrastive gap sizes is low between CUAXU and default loss at 32D (showed in Figure 3b). This shows that the representational spaces are more similar for 32D than for higher dimensionalities, and therefore there is a smaller performance difference at 32D.
> > >
> > > (We showed in Figure 3 that, as CLIP dimensionality increases from 32D to 512D, the difference in the sizes of the contrastive gap becomes more pronounced between CUA/CUAXU and default CLIP.)
> > >
> > > - **Small performance difference at 512D**: We suggest this is because the original CLIP model from which we fine-tuned our models was originally pre-trained at 512D. This might unfairly favor the original CLIP loss at this dimensionality, as the model learned to associate image-text pairs at 512D for a vast number of image-caption pairs from the internet. However, there is still a slight improvement in CUAXU's performance at 512D over default CLIP loss, showing the effectiveness of our method for average image-text retrieval.
> > >
> > >
> > > **Regarding larger-scale experiments**
> > >
> > > At the reviewer's suggestion, we fine-tuned CUAXU and default CLIP model on the much larger Conceptual Captions [1] dataset, containing about 2.2 million image-caption pairs (compared to 118k image-captions pairs in MS COCO). We added the results of this experiment in Figure 10 in Appendix A.5 in the updated paper.  We found that our proposed CUAXU loss outperforms default CLIP loss across all the three metrics we measured. We believe this shows a strong proof-of-concept of our approach, as we showed performance gains in the three evaluation metrics after fine-tuning on both small and large image-caption datasets.
> > >
> > > Finally, we thank the reviewer for pointing out the difficulty in identifying the changes in the updated paper. We highlighted the changes in blue in the new version.
> > >
> > > We kindly ask the reviewer to reassess their evaluation if these new updates address their concerns about our method's effectiveness.
> > >
> > >
> > > [1] Sharma, P., Ding, N., Goodman, S., & Soricut, R. (2018). Conceptual Captions: A Cleaned, Hypernymed, Image Alt-text Dataset For Automatic Image Captioning. Proceedings of ACL.

---

### Official Review · Reviewer_UiMt · 2024-11-12

**Soundness:** 2
**Presentation:** 2
**Contribution:** 2
**Rating:** 5
**Confidence:** 4

**Summary:**

An empirical study on the impact of several loss functions on the CLIP embedding space has been presented in this work. The experiment tries to investigate the modality gap between images and text in retrieval and classification tasks.

**Strengths:**

- A comprehensive loss analysis and introduction have been provided for the CLIP embedding space.
- The experiment is well-designed to demonstrate the *contrastive gap* in terms of several gap metrics and utility (e.g., retrieval accuracy).

**Weaknesses:**

- **Unclear Definition**: The proposed *contrastive gap* lies at the core of this work; however, it has never been defined clearly. While an intuitive example on the "idealized" dataset was given to demonstrate the concept, the setting of this example is less convincing (see the detailed comments below), and a clear, formal definition for the contrastive gap is still lacking.
- **Contrastive Gap v.s. Margin**: The experiment settings of `Table 1` and `Section 3.2` are unconvincing in investigating the modality gap and somewhat confused with the margin in a single image modality. As shown in Eq (2), the CLIP contrastive loss may work as a triplet loss to push the margin between positive and negative pairs. Thus, by using the same modality on both encoders, the learned contrastive gap is more like a margin among different samples instead of showing the modality gap.
- **Lack of Technical Novelty**: The key contribution of this work is relatively marginal. First, the proposed contrastive gap lacks insightful theoretical evidence and guarantees. Second, the proposed mitigation strategies all build on top of existing works.
- **Experiments**: While the new proposed fine-tuning loss shows some improvements, the CLIP embeddings badly degrade in retrieval, raising a severe concern about the utility of closing the "contrastive gap".

**Questions:**

- What's the formal definition of a contrastive gap?
- Is there any theoretical guarantee that closing the contrastive gap can bridge the modality gap?
- What's the difference between a contrastive gap and a margin?
- Can the proposed loss improve other common tasks, such as image captioning and text2image retrieval?

---

> ### Author Response · Authors · 2024-11-22
> **Response to Reviewer UiMt**
>
> We thank the reviewer for their helpful review. We first answer their questions, and then address the identified weaknesses below.
>
>
> ### Answers to Questions:
>
> **What's the formal definition of a contrastive gap?**
>
> The contrastive gap refers to the gap in the representational space between embeddings from two encoders trained with a multimodal contrastive loss (as defined in Equation (1) in the paper). This phenomenon is equivalent to the modality gap described in prior work [1], but we use the term contrastive gap to be more relevant when only one modality is present in both encoders in the model, as explored in our Section 3.2 experiments.
>
> **Is there any theoretical guarantee that closing the contrastive gap can bridge the modality gap?**
>
> Since we rename the modality gap as the contrastive gap in this work, closing the contrastive gap means the same thing as closing the modality gap.
>
> **What's the difference between a contrastive gap and a margin?**
>
> We use the term margin to refer to the same margin as in standard triplet loss:
>
> $L=\text{max}(s(a,n)−s(a,p)+\text{margin},0)$, where
>
> $s$ is the similarity function (Ex: cosine similarity),
> $a$ is the anchor,
> $p$ is the positive example, and
> $n$ is the negative example
>
> *In the multimodal setting*, the margin quantifies the separation between positive and negative image-text pairs.
>
> *The contrastive gap*, on the other hand, refers to the distinct geometric separation between the image cluster and text cluster in the embedding space.
>
> **Can the proposed loss improve other common tasks, such as image captioning and text2image retrieval?**
>
> We evaluated our method on ECCV-caption [2], which takes into account the presence of false negatives in MS COCO. We added this result to Figure 4 in the paper. We find that our method generally outperforms the default CLIP baseline on the retrieval task on ECCV-caption.
>
>
> ### Response to Weaknesses
>
> **Unclear Definition:**
> We clarified the definition of the contrastive gap in our response to Question 1. We acknowledge the need for a formal definition, and will add this to the revised paper.
>
> **Contrastive Gap v.s. Margin:**
> We showed in Eqn (2) that the CLIP loss may act as a triplet loss with zero margin.
> However, the contrastive gap specifically quantifies the separation between clusters of embeddings from encoder 1 and encoder 2 (which happen to be images in Section 3.2), not the margin between individual positive and negative pairs.
>
> **Lack of Technical Novelty:**
>
> We show the modality gap phenomenon persisting even in the presence of overlapping encoders, no mismatched pairs, and no information gap between modalities (i.e, both modalities are images). To our knowledge no other work has shown this. Further, while prior works mainly focus on *measuring* uniformity and alignment, we analyze the effect of directly *optimizing* uniformity/alignment in the multimodal contrastive learning setting.
>
> **Experiments:**
>
> As mentioned above, we added new results that show that our proposed method improves upon the baseline in the ECCV-captions evaluation.
>
> [1] Liang, Victor Weixin, et al. "Mind the gap: Understanding the modality gap in multi-modal contrastive representation learning." Advances in Neural Information Processing Systems 35 (2022): 17612-17625.
>
> [2] ECCV Caption: Correcting False Negatives by Collecting Machine-and-Human-verified Image-Caption Associations for MS-COCO, Chun et al., ECCV, 2022

---

> > ### Comment · Reviewer_UiMt · 2024-12-01
> > **Post-Rebuttal Feedback**
> >
> > Thanks for clarifying my previous questions. The reviewer appreciates the provided results on the ECCV-caption dataset. While it is interesting to understand the modality gap from the loss level, the main concerns still remain: 1) It is unclear if the contrastive gap can be disentangled with the margin between positive-negative pairs since both encoders employ the same-modal images. Additionally, will a similar contrastive gap persist when using the same-modal text data? 2) The experimental results on image-text retrieval are unconvincing in supporting the effectiveness of the proposed contrastive gap. The proposed loss cannot consistently improve the original CLIP loss and reports a similar performance when $d=512$. To sum up, the theoretical explanation of the proposed contrastive gap is still not convincing, and its effectiveness in bridging the modality gap is not well supported by empirical evidence. Thus, the reviewer will maintain the score unchanged.

---

> ### Author Response · Authors · 2024-12-02
> **Response to Reviewer UiMT's feedback**
>
> We thank the reviewer for their follow up. We address their concerns below:
>
> 1. We showed in Eqn (2) that the CLIP loss (at the same low temperature setting we used in Section 3.2), behaves as a triplet loss with zero margin. Further, to measure contrastive gap, we computed the centroid of images from encoder 1 as ```image_centroid1```, and the centroid of images from encoder 2 as ```image_centroid2```. Then, we compute contrastive gap as the distance between ```image_centroid1``` and ```image_centroid2```. Thus, we know that the **contrastive gap (distance between embeddings of each encoder) calculated is distinct from the margin of positive-negative pairs**, even though we conducted the experiment with only a single modality of images.
>
> - Further, we have the results for the same experiment in Section 3.2, when using the **same-modal text data**. We show those results below:
>
> |                          | At initialization | After 25 epochs |   |   |
> |--------------------------|-------------------|-----------------|---|---|
> | Centroid distance        | 0.01              | 0.57            |   |   |
> | Linear Separability Acc. | 0.44              | 1.00            |   |   |
> | Contrastive loss         | 0.03              | 0.00            |   |   |
>
>
> - We agree with the reviewer that the text-only results are also needed to paint the full picture, and will add these results to the final paper.
>
> 2. For image-text retrieval, we found that our proposed loss (CUAXU) improves over other baselines, but the improvement is marginal for 512D CLIP, and CUAXU performs similar to other baselines for 32D CLIP. We explain these results as follows:
>
> - **Small performance difference at 512D**: We suggest this is because the original CLIP model from which we fine-tuned our models was originally pre-trained at 512D. This might **unfairly favor the original CLIP loss at this dimensionality**, as the model learned to associate image-text pairs at 512D for a vast number of image-caption pairs from the internet. However, there is still a **slight improvement in CUAXU's performance at 512D over default CLIP loss**, showing the effectiveness of our method for average image-text retrieval.
>
> - **Small performance difference at 32D:** We suggest this is because the **difference between the contrastive gap sizes is low between CUAXU and default loss at 32D** (showed in Figure 3b). This shows that the representational spaces are more similar for 32D than for higher dimensionalities, and therefore there is a smaller performance difference at 32D.
>
> We will add these discussions to the final paper.
>
>
> Finally, we showed in Figure 3 that both our proposed losses (CUAXU and CUA) significantly reduced the modality gap in CLIP space: At 512D, **Linear Separability Accuracy is reduced by 22%** and **distance between image and text centers is reduced by more than 1000%** over default CLIP. Distance between image and text centers was the proposed metric to measure the modality gap by the paper [1] that introduced it, and linear separability accuracy is another well-established metric to measure the modality gap (used in [2, 3]).
>
> We kindly ask the reviewer to reassess their evaluation if these new updates and discussions address their concerns about our analysis of the contrastive gap, and our method's effectiveness.
>
> [1] W. Liang, Y. Zhang, Y. Kwon, S. Yeung, and J. Zou, ‘Mind the Gap: Understanding the Modality Gap in Multi-modal Contrastive Representation Learning’, arXiv [cs.CL]. 2022.
>
> [2] P. Shi, M. C. Welle, M. Björkman, and D. Kragic, ‘Towards understanding the modality gap in CLIP’, in ICLR 2023 Workshop on Multimodal Representation Learning: Perks and Pitfalls, 2023.
>
> [3] M. Al-Jaff, ‘Messing With The Gap: On The Modality Gap Phenomenon In Multimodal Contrastive Representation Learning’, Dissertation, 2023.

---

> > ### Comment · Reviewer_UiMt · 2024-12-03
> > **Post-Rebuttal Feedback Follow-up**
> >
> > Thanks for the further clarification. However, the limited image-text retrieval performance still casts a shadow on the effectiveness of the contrastive loss, especially in the case that the modality gap has been effectively reduced by using the proposed loss. This actually exacerbates my previous concerns about whether the proposed loss can disentangle the contrastive gap and the margin of positive-negative pairs. It seems like the proposed method closes the modality gap by sacrificing the margin.
> >
> > That being said, the reviewer appreciates the great efforts the authors have made in presenting the work. It would be helpful to compare the distribution or averaged values between the modality gap and the margin of all the positive-negative pairs. I will raise my score to 5 as the authors have clearly demonstrated the difference between the gap and margin, even though no further empirical evidence was provided due to the limited rebuttal time. However, my main concern about the utility of reducing the contrastive gap still persists.

---

> ### Author Response · Authors · 2024-12-04
> **Comparing Margin with Modality Gap**
>
> We thank the reviewer for their helpful feedback, and sharing their concerns. The hypothesis that our proposed method closes the modality gap by sacrificing margin is insightful. We also thank the reviewer for their suggestion to compare averaged values of margin and modality gap. We conducted a new experiment to determine the validity of the reviewer's hypothesis.
>
>
> ## Measuring Average Margin
> We measured the average margin between positive and negative image-text pairs as:
>
>
> - Average margin = $D_n - D_p$, where,
> - $D_n$ = Mean distance between negative image-text pairs.
> - $D_p$ = Mean distance between positive image-text pairs.
>
> Distance here can either be euclidean or cosine.
>
> We measured the margin in 512D CLIP, and found that the **margin is higher for models finetuned with our proposed losses** and the **contrastive gap is lower**, suggesting that **our proposed method does not sacrifice margin to close the modality gap**. We confirm that this is true for both euclidean and cosine measures of distance.
>
> ## Margins
>
>
> | Model                 | Margin (Mean Pairwise Euclidean Distance) | Margin (Mean Pairwise Cosine Distance) |
> |-----------------------|-------------------------------------------|----------------------------------------|
> | **OpenAI-Pretrained** | 0.12                                      | 0.15                                   |
> | **Default CLIP loss** | 0.18                                      | 0.24                                   |
> | **CUAXU loss**        | 0.43                                      | 0.51                                   |
> | **CUA loss**          | **0.54**                                  | **0.62**                               |
>
> - We found that our proposed losses do not sacrifice margin to close the modality gap. In fact, our proposed losses increase the margin relative to the baselines, indicating that there is better separation between positive and negative image-text pairs.
>
>
>
> ## Modality Gap
>
>
> | Model                 | Modality Gap (Distance between centroids) | Modality Gap (Mean Pairwise Cosine Distance) |
> |-----------------------|-------------------------------------------|----------------------------------------------|
> | **OpenAI-Pretrained** | 0.82                                      | 0.70                                         |
> | **Default CLIP loss** | 0.68                                      | 0.74                                         |
> | **CUAXU loss**        | 0.12                                      | 0.49                                         |
> | **CUA loss**          | **0.07**                                  | **0.38**                                     |
>
> - We confirm that our proposed losses result in lower modality gap in terms of both cosine and euclidean distances. Lower modality gap results in positive image-text pairs being closer together in the representational space.
>
> These new results show that our proposed methods bridge the modality gap by bringing positive image-text pairs closer together, while increasing the distance between negative image-text pairs beyond that of the compared baselines.
>
> Finally, we highlight that our proposed methods show increased performance in not just the image-text retrieval task, but also zero-shot image classification and multi-modal arithmetic tasks over the baselines. These results provide evidence of the utility of reducing the contrastive gap.

---

### Author Response · Authors · 2024-11-22
**General Response (1)**

We thank all the reviewers for their insightful comments and suggestions on our work. We agree that incorporating their feedback will improve the overall quality of our work.

In this post, we outline:
1. The positive feedback from the reviewers.
2. The results of our new experiment showing improved image-text retrieval performance of our method.

### 1. Reviewers’ Positive Feedback

We are encouraged by the reviewer’s positive feedback on the paper. Reviewers found:

**Our contributions significant**:

- ```UiMt```: “A comprehensive loss analysis and introduction have been provided...”
- ```Mgbo```: “finding that a gap also arises within the same modality is significant and could impact a broad range of applications.”
- ```pp1S```: “ provides a comprehensive empirical analysis...offering insights into why this phenomenon occurs in multi-modal models.”

**Our experiments well-designed:**

- ```UiMt```: “The experiment is well-designed to demonstrate the contrastive gap in terms of several gap metrics and utility”
- ```pp1S```: “The experiments are extensive, covering multiple evaluation metrics and downstream tasks”
- ```iaqR```: “Simple experiments prove the motivation.”

**Our ideas clear and easy to follow**:
- ```Mgbo```: “The analysis of the rationale behind how contrastive loss induces a gap is clear and reasonable.”, “The paper is well-written and easy to follow.”
- ```iaqR```: “The paper is clearly written and the method can be well understood.”

**Our method simple to implement:**
- ```pp1S```: “The proposed solution ... is relatively simple to implement”

### 2. New image-text retrieval performance result

Here, we address the concern about the lower retrieval performance of our proposed method on the MS COCO dataset. In the paper, we suggested that this could be due to the presence of false negatives in MS COCO (Discussed in the paper in Appendix Section A.3). At the suggestion of reviewer ```Mgbo```, we evaluated our proposed methods on ECCV-caption [1] to provide a stronger demonstration of our method's effectiveness.

We find that CLIP fine-tuned with our proposed $\text{L}_{\text{CUAXU}}$ generally outperforms default fine-tuned CLIP on the retrieval task on ECCV-caption, supporting our discussion in Section A.3. We updated the paper to show the results of this evaluation in Figure 4.

We reported the mAP@R metric which is more aligned to humans than Recall@K, and is therefore recommended by Chun et al [1]. We also updated the text in Sections 5.2 and 5.3 to reflect this change. For now, we added the models fine-tuned with default CLIP loss and $\text{L}_{\text{CUAXU}}$ to show proof-of-concept. We plan to add the rest of the baselines to Figure 4 before the discussion window closes.

[1] ECCV Caption: Correcting False Negatives by Collecting Machine-and-Human-verified Image-Caption Associations for MS-COCO, Chun et al., ECCV, 2022

---

> ### Author Response · Authors · 2024-11-28
> **General Response (2)**
>
> We updated our PDF (changes marked in blue), and outline the major changes in this post:
>
> 1. The baselines we added to our ECCV caption evaluation
> 2. Results of fine-tuning CLIP models on larger image-caption dataset
> 3. Added OpenAI's pretrained CLIP model as a baseline in our experiments
>
> ### 1. Results for ECCV Evaluation
>
> We added the rest of the baselines in Figure 4, showing that models fine-tuned with CUAXU loss beat the other baselines. However, we found that models fine-tuned with CUA loss perform worse than models fine-tuned with default CLIP loss in the image-text retrieval task. This result implies that a reduced contrastive gap (in the case of CUAXU loss) may lead to more accurate image-text correspondences. However, it is important to optimize for cross-modality uniformity, and therefore drive negative image-text pairs further apart, in order to perform well in the image-text retrieval task (as evidenced by the relatively poor performance of CUA loss in this task). Furthermore, we saw the importance of the cross-modality uniformity term in zero-shot transfer performance as well (Figure 5 in the paper), where models fine-tuned with CUAXU loss outperform those fine-tuned with CUA loss.
>
> ### 2. Results for fine-tuning CLIP models on larger dataset
>
> In order to provide a stronger proof-of-concept of our approach, we evaluated our proposed method on the significantly larger conceptual captions dataset [1]. In our experiment, we were able to access about 2.2 million image-caption pairs through image urls specified by the dataset. We fine-tuned two models, one with default CLIP loss, and another with our proposed CUAXU loss. We added these results in a bar plot in Figure 10 in Appendix A.5 in the updated paper. We found that the CLIP model fine-tuned with CUAXU loss outperforms the model fine-tuned with default CLIP loss across all the three metrics we measured.
>
> [1] Sharma, P., Ding, N., Goodman, S., & Soricut, R. (2018). Conceptual Captions: A Cleaned, Hypernymed, Image Alt-text Dataset For Automatic Image Captioning. Proceedings of ACL.
>
> ### 3. OpenAI's pretrained CLIP model as a baseline
>
> Following reviewer ```pp1S```'s suggestion, we added OpenAI's pretrained CLIP model (untuned CLIP) as a baseline in our experiments in Figures 3, 4, 5, and 6. We show that our proposed losses **reduce the contrastive gap over OpenAI's pretrained CLIP**. Furthermore, our proposed losses **improve performance over OpenAI's pretrained CLIP in average image-text retrieval (CUAXU loss), and multimodal-arithmetic (both CUA and CUAXU loss)**. We provide a discussion of the comparison of the zero-shot performances in Section 5.3 in the paper.

---

### Meta-Review · Area_Chair_hgvZ · 2024-12-20

**Metareview:**

This paper investigates the "modality gap", which is a widely known phenomenon in CLIP embedding space. This paper proposes to rename "modality gap" to "contrastive gap" because the gap emerges by contrastive training rather than modality difference. This paper also shows that contrastive gap can be more significant when using small batch sizes in a high-dimensional space. To tackle the problem, this paper proposes to add uniformity and alignment losses and evaluates the fine-tuned models on various downstream tasks.

This paper has mixed opinions, mostly negative ones. The reviewers share the following main concerns:

- [UiMt, pp1S] Insufficient justification for using "contrastive gap" rather than "modality gap". Specifically, Reviewer pp1S pointed out that the main claim of this submission is built upon a misunderstanding of the CLIP loss function and Wang & Isola (2020). Similarly, the relationship between reducing contrastive gap and performance improvement has no sufficient justification.
- [UiMt, Mgbo, iaqR] The fine-tuned CLIP by the proposed method shows a degraded retrieval performance. It does not support well the main claim that bridging the contrastive similarity gap benefits representation learning.
- [Mgbo, pp1S] Experiments are only conducted on a small-scale fine-tuning, which can lead to a different result for a general from-scratch CLIP training.
- [UiMt, Mgbo, iaqR] Lack of technical novelty. For example, this submission is closely related to (Al-jaff, 2023), or alignment and uniformity losses are often used by CLIP training.

I also agree with the reviewers' opinions. Specifically, as pointed out by Reviewer Mgbo and pp1S, the current study is somewhat limited because all the experiments are based on fine-tuning on small-size dataset. Although the additional ECCV Caption evaluation justifies the contradictory result in the retrieval tasks, this paper still has large room for improvements, for example, the lack of the from-scratch training (I don't think fine-tuning with larger image-text pairs can replace from-scratch training experiments) and somewhat insufficient explanation about "contrastive gap" (e.g., why we have to rename modality gap to contrastive gap?).

**Additional Comments On Reviewer Discussion:**

While all the reviewers with negative opinions engaged in the discussion and kept the negative opinion, the reviewer with a positive opinion did not engage in the discussion despite reminders by the authors and the AC.

The authors provide additional experimental results on ECCV Caption to support the fine-tuned CLIP performs well on retrieval tasks. All the reviewers engaged in the discussion satisfy with this response.

Also, the authors provided additional experiments with a larger image-text dataset, but both Reviewer Mgbo, pp1S expected from-scratch training results. As mentioned in my previous comment, I also think that this result cannot replace the from-scratch training result.

---

### Decision · Program_Chairs · 2025-01-22

Reject